# Effect of friction on oxidative graphite intercalation and high-quality graphene formation

Steffen Seiler[1], Christian E. Halbig[2,3], Fabian Grote [2,3], Philipp Rietsch[2,3], Felix Börrnert[4], Ute Kaiser[4], Bernd Meyer [1] & Siegfried Eigler[2,3]

Oxidative wet-chemical delamination of graphene from graphite is expected to become a scalable production method. However, the formation process of the intermediate stage-1 graphite sulfate by sulfuric acid intercalation and its subsequent oxidation are poorly understood and lattice defect formation must be avoided. Here, we demonstrate film formation of micrometer-sized graphene flakes with lattice defects down to 0.02% and visualize the carbon lattice by transmission electron microscopy at atomic resolution. Interestingly, we find that only well-ordered, highly crystalline graphite delaminates into oxo-functionalized graphene, whereas other graphite grades do not form a proper stage-1 intercalate and revert back to graphite upon hydrolysis. Ab initio molecular dynamics simulations show that ideal stacking and electronic oxidation of the graphite layers significantly reduce the friction of the moving sulfuric acid molecules, thereby facilitating intercalation. Furthermore, the evaluation of the stability of oxo-species in graphite sulfate supports an oxidation mechanism that obviates intercalation of the oxidant.

[1] Interdisciplinary Center for Molecular Materials (ICMM) and Computer-Chemistry-Center (CCC), Friedrich-Alexander-Universität Erlangen-Nürnberg (FAU), Nägelsbachstraße 25, 91052 Erlangen, Germany. [2] Department of Chemistry and Pharmacy and Institute of Advanced Materials and Processes (ZMP), Friedrich-Alexander-Universität Erlangen-Nürnberg (FAU), Henkestraße 42, 91054 Erlangen, Germany. [3] Institute of Chemistry and Biochemistry, Freie Universität Berlin, Takustraße 3, 14195 Berlin, Germany. [4] Materialwissenschaftliche Elektronenmikroskopie, Universität Ulm, Albert-Einstein-Allee 11, 89081 Ulm, Germany. Correspondence and requests for materials should be addressed to B.M. (email: bernd.meyer@chemie.uni-erlangen.de) or to S.E. (email: siegfried.eigler@fu-berlin.de)

The outstanding electronic and mechanic properties of graphene and its chemical reactivity led to the development of graphene technology[1,2]. However, mainly high temperature processes are facilitated to produce graphene[3–5]. Up to now, it remains challenging to prepare the same quality of graphene wet-chemically utilizing chemical functionalization methods. The advantages of a wet-chemical process are its scalability and the possibility to introduce functional groups at the same time. Chemical modification can give new functions to graphene, which enable e.g., medical applications[2,6–10].

Conventional synthesis of graphene via graphite intercalation, oxidation, delamination into graphene oxide, and subsequent reduction (see Fig. 1a for a schematic illustration of the process), however, leads in most cases to a ruptured carbon lattice[11]. For improving the quality of the produced graphene, it is therefore highly desired to understand the underlying mechanisms in this process in more detail in order to be able to better control the individual synthesis steps. Key factors on the way to high-quality graphene are the proper formation of the intermediate graphite sulfate, a stage-1 graphite intercalation compound (GIC) with every layer of the graphite intercalated (see Fig. 1), and a controlled oxidation of the graphite sulfate[12]. Concerning the graphite oxidation, significant progress was recently made by adjusting the reaction conditions[13,14]. We demonstrated the wet-chemical oxidative delamination of graphite forming a derivative of graphene functionalized by oxo-addends with lattice defect concentrations ($\theta_D$) of 0.3–0.05%[13,14]. This oxo-functionalized graphene was found to be suitable to establish a controlled chemistry of graphene where lattice defects play a minor role[12,15–18]. We term this product oxo-$G_1$ (index = number of layers) in order to distinguish it from the known graphene oxide, which possesses an undefined amount of defects and consequently a disturbed order of the lattice.

Nevertheless, how a proper stage-1 GIC is obtained that enables oxo-$G_1$ formation (Fig. 1a) and the oxidation process itself are not well understood. The formation and the properties of the graphite sulfate intercalate, for which the ideal formula of $(C_{24}^+(HSO_4^-)(H_2SO_4)_2)_n$ has been reported, needs more clarification for improving the yield and the quality of the synthesized graphene[19–21]. The intercalation of graphite with sulfuric acid increases the layer distance to 7.98 Å[22–24]. This intercalation process is initiated by electronic oxidation of the graphite[20,23,25–29]. However, why the initial oxidation is needed

and to what extend oxo-functional groups are introduced on the basal planes of the graphite scaffold at this stage, is currently unknown. Puzzling are also recent contradictory reports: while some groups observed the intercalation process to be reversible upon water addition[21,24], we recently reported the formation of hydroxylated graphene under similar reaction conditions[14]. This contradiction unfolds the lack of our understanding of the overall graphene synthesis reaction.

Here we reveal that crystallinity of the graphite is the decisive factor for obtaining a proper stage-1 GIC and subsequently oxo-$G_1$. Our oxo-$G_1$ from the graphite with the highest crystallinity yields a graphene with an exceptionally low density of lattice defects down to 0.02% after reductive defunctionalization. Statistical Raman spectroscopy (SRS) and high-resolution transmission electron microscopy (HRTEM) prove the high lattice quality. Mechanistic ideas about the formation ability of different types of graphites to form stage-1 graphite sulfate are derived from ab initio molecular dynamics (MD) simulations, which provide an understanding why a high degree of crystallinity of graphite is mandatory. The simulations show that the friction of moving sulfuric acid molecules in the graphite intercalation compound is intimately coupled to the stacking and the electronic oxidation state of the graphite layers. Ideal AB stacking and initial oxidation significantly reduces friction and supports the spreading of the sulfuric acid molecules within the graphite. Oxo-species (epoxides, OH groups) are found to be unstable within the graphite intercalation compound below a formal oxidation state of about $C_{30}^+$. Therefore, it is not required that oxidation takes place within the layers of graphite by intercalation of the oxidant, but can proceed from the outside via electron transfer from the conducting graphite layers and charge compensation by Grotthuss proton diffusion in the sulfuric acid network as illustrated in Fig. 1b.

## Results

**Graphite intercalation and graphene preparation.** The formation of graphite sulfate was studied experimentally by immersing a high quality graphite crystal in a quartz glass cuvette with conc. sulfuric acid and ammonium persulfate as oxidant (Fig. 2). The formation of stage-1 graphite sulfate was indicated by the blue color (Fig. 2b). However, also mixed stages of graphite sulfate are formed, indicating that stage-1 graphite sulfate formation is hindered in some places (variable bluish and greenish colors—Fig. 2b, c). Although the facet-like $H_2SO_4$-GIC formation was

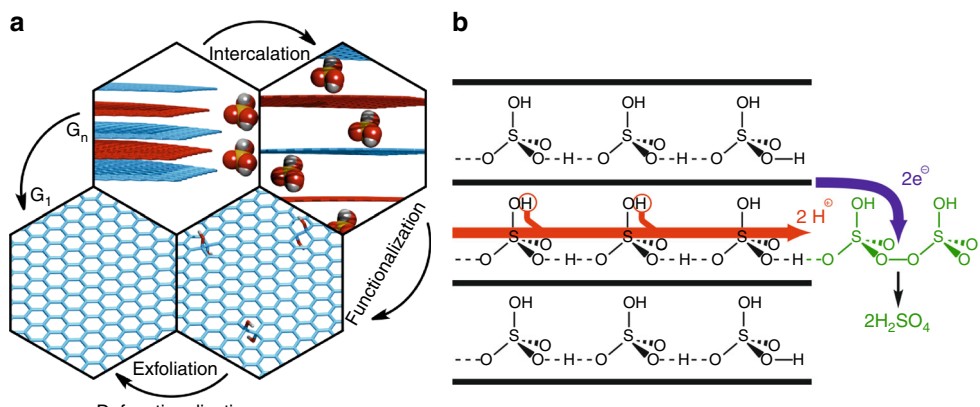

**Fig. 1** Mechanism of graphene oxidation and exfoliation. **a** Schematic illustration of the chemical process of graphene formation. Interaction of graphite with sulfuric acid and an oxidizing agent leads to oxo-functionalized graphene after hydrolysis and enables delamination. Graphene is formed by reductive defunctionalization ($G_n$: graphite, $G_1$: graphene, index: number of layers). **b** Sketch of the proposed oxidation mechanism: electrons (blue) are transferred from conducting graphite sheets to the oxidizing agent, while charge balance is maintained by proton hopping through the hydrogen bond network of the sulfuric acid

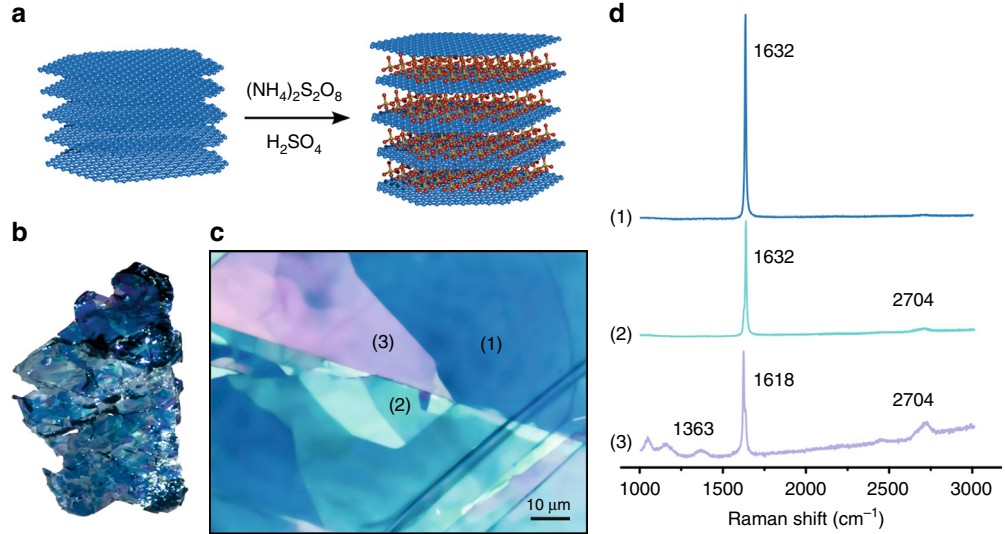

**Fig. 2** Intercalation of highly crystalline graphite. **a** Illustration of the reaction of ideally AB-stacked graphite with ammonium persulfate in sulfuric acid forming stage-1 graphite sulfate. **b** Optical image of a treated natural flake of graphite for one month and **c** microscopy image with dark blue areas (1) that can be related to stage-1 graphite sulfate and other regions lighter in color indicating that stage-1 formation is hindered (2,3). **d** Raman spectra of area 1, 2, and 3. See also Supplementary Methods

described in the literature[21], the origin of hindrance could not yet be elucidated.

To shine more light on the intercalation and subsequent functionalization and graphene formation process three different grades of graphite were used: Sri Lankan natural graphite with 70–150 μm grain (NG1), natural graphite with 10 μm grain (NG2), and turbostratic graphite (TG). X-ray powder diffraction of those graphites was conducted in an earlier study[30], and NG1 is the most crystalline graphite with 28% 3R-phase (ABC stacking) and crystallite size larger than 10 μm and 72% 2H-phase (AB stacking) with crystallite size of about 100 nm. NG2 also bears crystallinity with about equal amounts of 3R-phase (46%) and 2H-phase (54%) with however crystallite sizes of 100 nm and less. In contrast, TG bears no significant stacking order[30]. Raman spectroscopy of those graphite grades (Fig. 3) reveals the average distance of lattice defects ($L_D$) according to the intensity ratio of the defect induced D peak and the G peak ($I_D/I_G$)[30–32], which are for NG1 $L_D = 35$ nm ($I_D/I_G = 0.12$), for NG2 $L_D = 36$ nm ($I_D/I_G = 0.11$) and for TG $L_D = 19$ nm ($I_D/I_G = 0.38$). The symmetrical shape of the 2D peak of TG also indicates the missing stacking order[30,33].

The intercalation experiments and Raman analyses reveal the successful stage-1 formation for NG1 and NG2, indicated by the G peak shift to 1630 cm$^{-1}$ due to the high doping level. The 2D peak is almost absent in both NG-GICs as expected (Fig. 3)[24]. In contrast, TG is not forming a proper stage-1 GIC and thus, next to the G-peak shift, the D and 2D peak are still present.

Subsequently, GICs were reacted with water and substantial changes of the Raman spectra are observed for all graphite grades. Most of the recorded Raman spectra of flakes of processed NG1 and NG2 are similar to the ones obtained for graphite oxide with comparable full-widths at half-maximum ($\Gamma$) of the D and G peak, respectively (NG1: $\Gamma_D/\Gamma_G = 89/81$ cm$^{-1}$; NG2: $\Gamma_D/\Gamma_G = 85/70$ cm$^{-1}$; TG: $\Gamma_D/\Gamma_G = 97/81$ cm$^{-1}$)[11]. However, few spectra of processed NG2 also exhibit characteristic spectral features of graphite (Fig. 3). In contrast, TG reverts back to the initial quality of graphite and single Raman spectra related to oxidized graphite are sporadically found. Such a reversibility of intercalation was also reported by Hofmann (described in 1938) and recently by Dimiev et al.[19,24]. Our results reveal that functionalization of graphite sulfate is possible, but the success is dependent on the graphite grade and the ability to form stage-1 graphite sulfate.

Subsequent sonication of water purified samples of functionalized graphite leads to exfoliation and a deposition of flakes on Si / 300 nm SiO$_2$ substrates followed by chemical reduction for further analysis becomes possible. Accordingly, a majority of flakes of graphene ($\Gamma_{2D} < 35$ cm$^{-1}$) with lateral dimensions up to 10 μm is found for NG1, whereas for NG2 such flakes are only isolated from the supernatant and with lateral dimensions <3 μm. Apparently, mainly graphite-like particles are formed. In contrast, graphene was not found from TG. Consequently, we prepared dispersions of functionalized graphene from NG1 to investigate the potential use of this synthetic approach.

**Graphene films and atomic lattice structure**. Although a larger fraction of the functionalized material remains few-layered and multi-layered, dispersions of oxo-G$_1$ with an optical density of 0.5 could be obtained by iterative centrifugation and redispersion in a 1/1 mixture of H$_2$O/MeOH assisted by ultrasonication (Supplementary Fig. 1). A strongly red shifted absorption maximum at 252 nm in comparison to 235 nm for oxo-G$_1$ with $\theta_f = 0.5$ ($\theta_f$: degree of functionalization) indicates a low degree of oxo-functionalization, which was estimated to $\theta_f = 0.04$ in an earlier study[14]. Furthermore, graphene obtained after reduction with NaBH$_4$ and stabilized by sodium cholate exhibits an absorption maximum at 266 nm, instead of 254 nm for oxo-G$_1$ ($\theta_f = 0.5$), indicating an improved quality of graphene (Fig. 4a).

Furthermore, the dispersions allow for film formation by Langmuir–Blodgett (LB) technique. In Fig. 4d an optical microscopy image illustrates partially overlapping flake areas with few thicker particles. The lateral dimensions of flakes are about 0.5–3 μm as depicted in the atomic force microscopy image of Fig. 4b. This film of graphene also allows determining the density of defects by statistical Raman spectroscopy (SRS). The average $I_D/I_G$ ratio is determined to $1.2 \pm 0.2$ and the $\Gamma_{2D}$ to $36 \pm 3$ cm$^{-1}$. Accordingly, $\theta_D$ is about 0.025%[31,32]. This value suggests areas with an intact graphene lattice of 4000 carbon atoms ($L_D = 11$ nm). Moreover, the $I_G$ values of recorded Raman data let conclude that about 72% of the area are either single layer graphene (including areas of probed single layer graphene / substrate) with only 28% of few-layer graphene (Supplementary Fig. 2). The average $I_D/I_G$ ratio of few- and multi-layer Raman

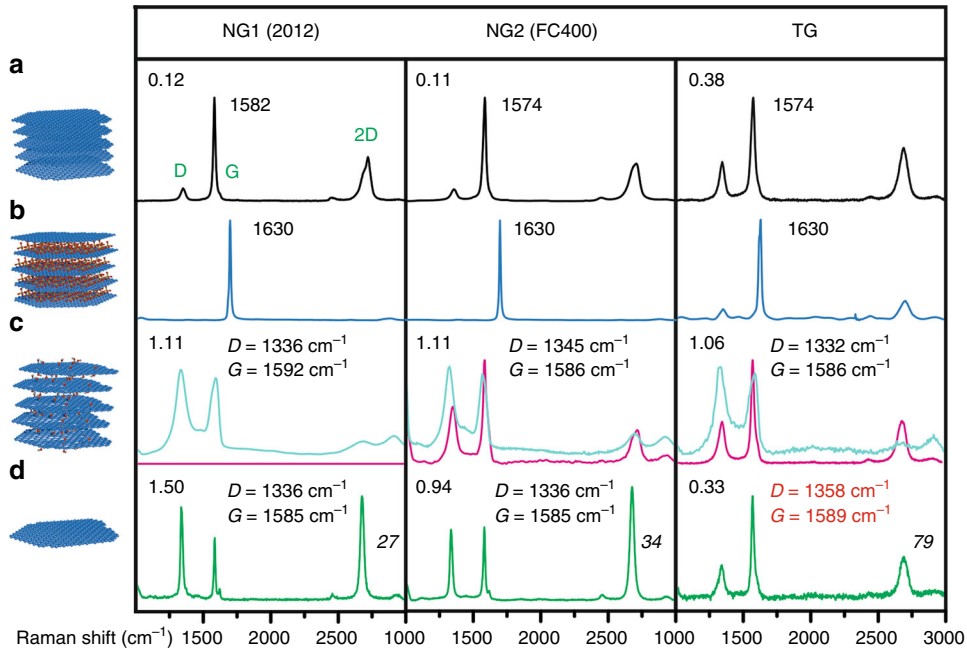

**Fig. 3** Raman study on the intercalation of natural graphite grades (NG1, NG2) and turbostratic graphite (TG). **a** Averaged Raman spectra of pristine graphites (black spectra). **b** Average spectra of graphites after treatment at intercalation reaction conditions to form stage-1 graphite sulfate (dark-blue spectra). **c** Extracted single spectra of recorded maps after water addition and sonication: Oxidized graphite (bright-blue) and unoxidized areas found in NG2 and TG (magenta). **d** Raman spectra of flakes of graphene from NG1 isolated by centrifugation, Raman spectra of graphene from NG2 from the supernatant, and Raman spectra from TG from smaller shards of graphite (green spectra). See also Supplementary Methods

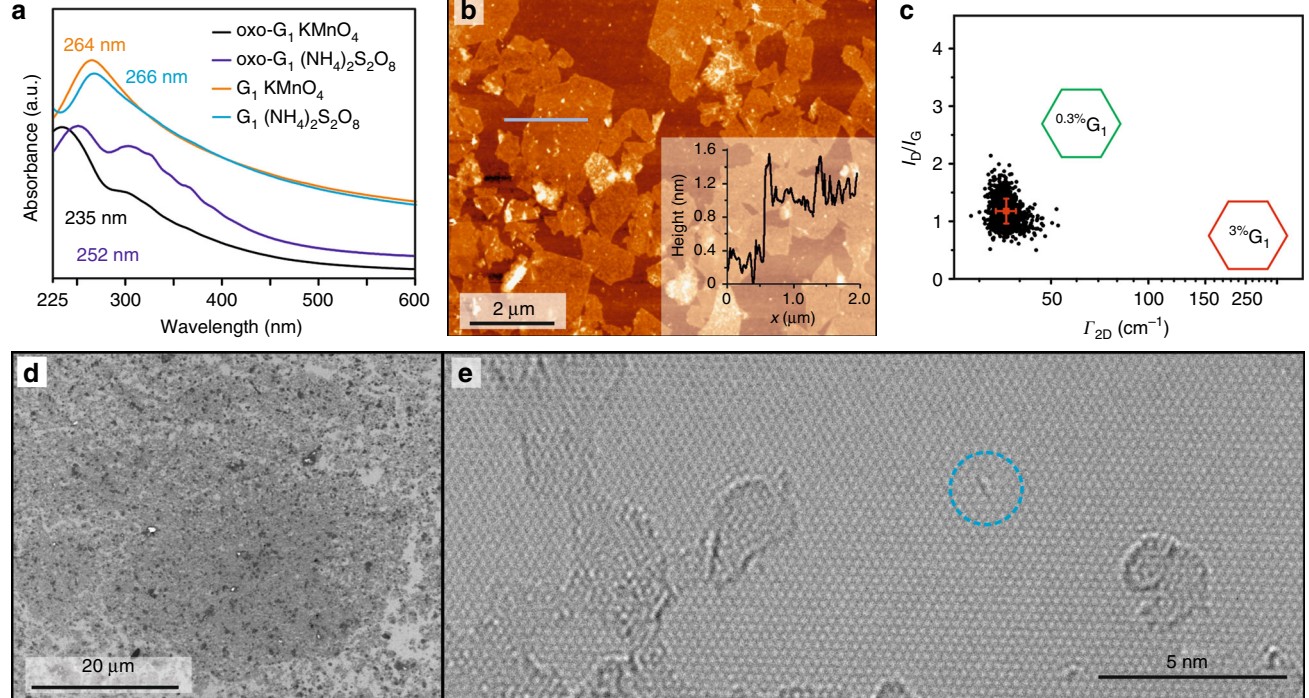

**Fig. 4** Characterization of the exfoliated graphene. **a** UV–Vis spectra of oxo-$G_1$ and their corresponding reduced species. **b** AFM image of the same film and height profile along gray line. **c** SRS results of the graphene film (density of defects about 0.025% (black)) compared to graphene derived from oxo-$G_1$ with a density of defects of 0.3% (green) and about 3% (red), respectively. **d** Optical microscope image of a film of flakes of graphene prepared by LB technique on 300 nm $SiO_2$ surface. **e** TEM image shows the resolved carbon lattice of graphene next to amorphous impurities; lattice defects are mainly point defects (for a large-scale image see Supplementary Figs. 4 and 5)

spectra is found to decrease, as a consequence of overlapping flakes with lower densities of defects, (Supplementary Fig. 3).

The interpretation of SRS results is further confirmed by HRTEM performed with atomic resolution after deposition of functionalized graphene on TEM grids by the LB technique. The carbon lattice was observed without additional cleaning or heating procedures, as we described in a recent HRTEM study[34]. Although heteroatoms, such as nitrogen or oxygen, can't be discriminated from carbon by contrast, the carbon framework of graphene is well resolved as it can be seen in Fig. 4e and Supplementary Figs. 4 and 5. The HRTEM image shows few larger holes within an almost intact graphene membrane, which is decorated with amorphous material—most probably carbon (according to the contrast) located on both sides of the layer. Amorphous impurities can originate from the reaction itself, or from ambient and wet-chemical processing. The diffractogram shows the six-fold symmetry of the lattice as sharp spots and the center spot originates from amorphous material. Magnification of the carbon framework (Fig. 4e) displays point-like defects, such as missing C atoms or a rearranged carbon lattice. Those types of defects have also been observed in graphene prepared by other methods[35]. Although some areas of the carbon lattice are hidden by the adsorbed material, the periodic extension of the uncontaminated carbon framework gives evidence for a coherent hexagonal lattice.

**Molecular dynamics simulations**. The observation that micron-sized graphite grains with perfect crystalline stacking (NG1) are easier to intercalate and to oxidize than more defective graphite grades with smaller grains (NG2) and perturbed stacking (TG) is surprising. Intuitively one would expect that the reduced cohesion between the graphite layers and the increased layer separation in perturbed graphite makes it easier for the sulfuric acid molecules to enter the graphite lattice, widen the layer spacing and fill the available space. This suggests that for GIC formation and oxidation the initial ability of the sulfuric acid to penetrate the graphite lattice is less important, but the mobility of the sulfuric acid between extended graphite sheets is the more decisive factor. To explore this idea, a series of ab initio molecular dynamics (MD) simulations were performed in order to determine the interaction of confined sulfuric acid molecules with the graphite lattice[36] and to probe how their mobility depends on the graphite stacking and the GIC oxidation state.

**Stability of oxo-species**. First, we investigated the stability of hydroxyl and epoxide groups attached to the graphite layers in the presence of a concentrated sulfuric acid medium in order to establish realistic structural models for the oxidized GIC. A periodic orthorhombic supercell was used with lateral extension of $5 \times 3\sqrt{3}$ graphene lattice constants (i.e., 60 carbon atoms) and an interlayer separation of 9.12 Å as suggested by experiment for partially oxidized stage-1 GIC[22]. The region between the graphite layers was filled with 8 sulfuric acid and one water molecules to reflect the composition of concentrated sulfuric acid (see Method section). Different stages of oxidation were obtained by adding one, two and three isolated pairs of hydroxyl groups to this unit cell, each in 1,2-*trans* configuration since this is the most stable arrangement of OH pairs (see Fig. 5a)[37].

Within a few picoseconds of simulation time, sporadic proton transfer events from sulfuric acid molecules to the covalently bound hydroxyl groups were observed. This finally led to the formation of water molecules, which detached from the carbon scaffold and entered the liquid layer. However, such events were only seen twice per simulation (even after long simulation times of more than 40 ps), leaving behind a two-fold positively charged graphite layer (see Fig. 5b). Similarly, when simulations were

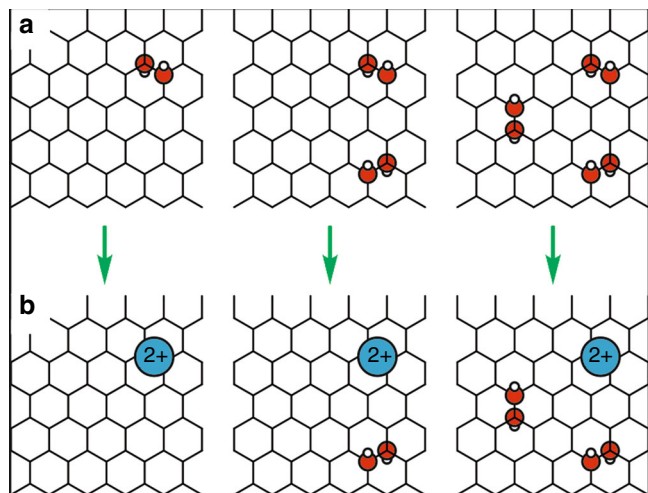

**Fig. 5** Sketch of spontaneous dehydroxylation events for different OH coverages. **a** Initial configurations, **b** result after 40 ps of ab initio MD

repeated with epoxides instead of hydroxyl groups, only the spontaneous conversion of one epoxide to water was observed, leading to the same formal charge of the graphite layers of +2 per supercell. To prove that the remaining oxo-species are indeed stable, we enforced the detachment of a third OH group and calculated the free energy cost by thermodynamic integration, for which we obtained about 60 kJ mol$^{-1}$ (see Supplementary Note 1).

The insertion of 2 positive charges per 60 carbon atoms by spontaneous protonation events matches very well the experimentally determined degree of oxidation of stage-1 GIC $(C_{24}^{+}(HSO_4^{-})(H_2SO_4)_2)_n)$ of one positive charge per 24 carbon atoms[19] and can be well understood from an analysis of the electronic structure (see Supplementary Note 2). In addition, the MD simulations show, in agreement with experiment, that hydroxyl groups are stable species at higher oxidation states and remain attached to the carbon scaffold, especially if the water content in the sulfuric acid is increased.

**Friction between liquid sulfuric acid and carbon layers**. To study the influence of graphite stacking and oxidation on the dynamics and the relative lateral movement of the intercalated sulfuric acid molecules we performed MD simulations for three different model configurations which we termed GIC-AA, ox-GIC($C_{30}^{+}$)-AA, and ox-GIC($C_{30}^{+}$)-AB (see Fig. 6a–d). First, two different stacking sequences of the graphite layers were considered: AB stacking as in the thermodynamically most stable hexagonal (ABAB) and rhombohedral (ABCABC) graphite modifications, and AA stacking, in which all C atoms sit directly on top of each other, as a representative of less favorable stackings occurring in randomly ordered graphites with no distinct stacking order, for example, turbostratic graphite (Fig. 6b, c). In addition, simulations were done for a neutral and an oxidized graphite with a formal charge of $C_{30}^{+}$. This choice of oxidation state was motivated by the results from our study on the stability of hydroxyl and epoxide groups described above. Accordingly, one H atom per 30 carbon atoms was removed from the sulfuric acid layer, resulting in $HSO_4^{-}$ species, which obtain their additional electron from an oxidation of the graphite sheets.

Inspired by the work of Tocci et al.[38], who studied the friction for liquid water gliding over graphene and boron nitride surfaces, we quantify the resistance for the liquid sulfuric acid to move between the solid graphite layers in the GIC by calculating the associated friction coefficient. A higher friction coefficient indicates a lower mobility and a lower transport of sulfuric acid

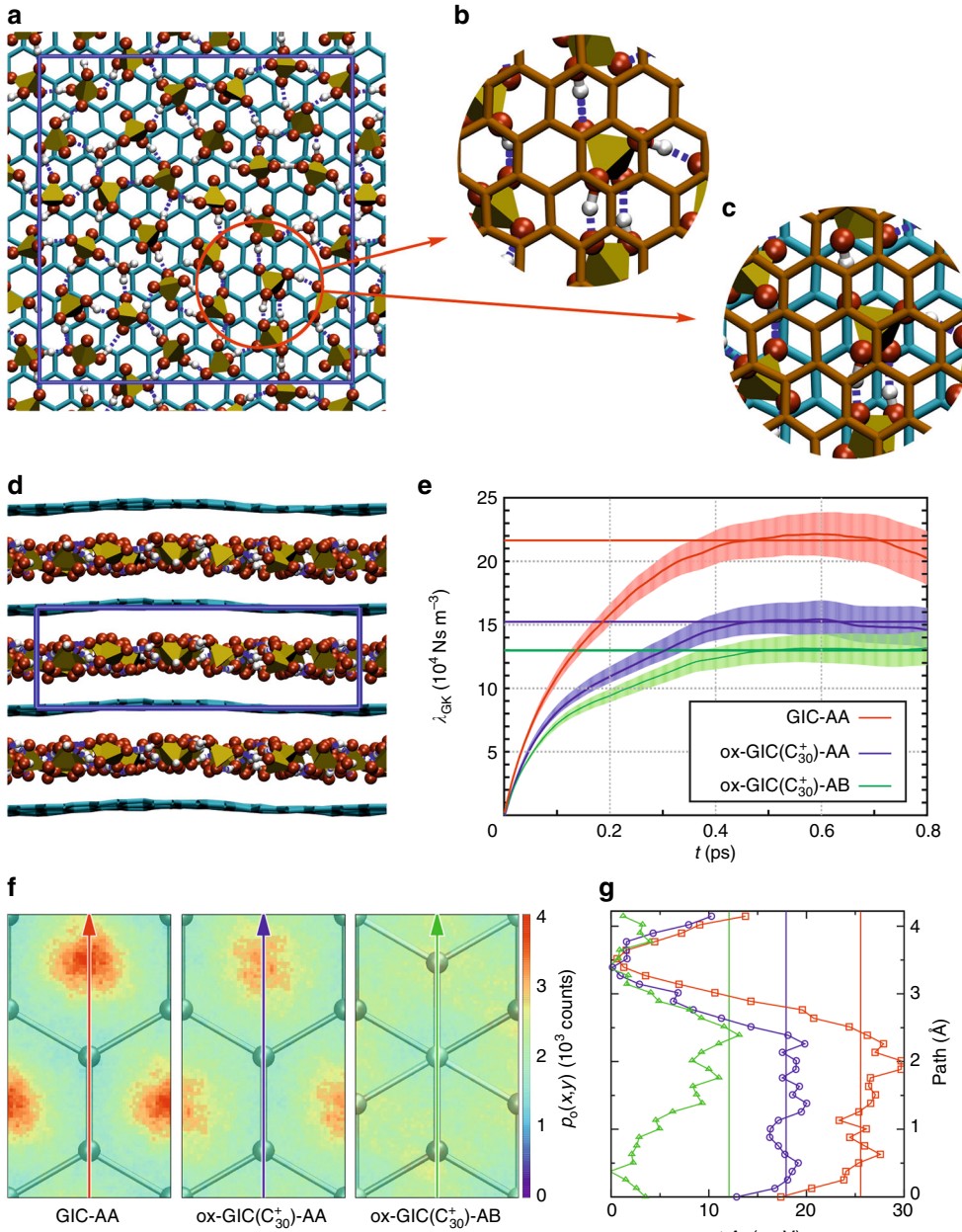

**Fig. 6** Molecular dynamics simulations. **a**, **d** Top and side view of the ox-GIC($C_{30}^+$)-AA supercell. **b**, **c** Enlargement of the top view including a second graphene layer (brown) for the ox-GIC($C_{30}^+$)-AA and the ox-GIC($C_{30}^+$)-AB simulation. Turquoise and brown: carbon framework, yellow: sulfate tetrahedron, red: oxygen atoms, gray: hydrogen atoms, blue: simulation box. **e** Green-Kubo friction coefficient as function of simulation time. Shaded areas: error bars represent standard error of the mean as obtained from block averaging. New starting points were set every 1000 MD time steps (see Supplementary Note 3). **f** Probability distribution map of oxygen atoms (same z-scale for all subfigures) mapped on a single orthorhombic unit cell. **g** Line scan through oxygen distribution map along the red, blue and green arrows

with respect to the graphite scaffold. The friction coefficient $\lambda$ is defined as the ratio between the friction force parallel to the carbon sheets $\mathbf{F}_P$ per unit area $A$ and the velocity jump $v_{slip}$ at the interface[38]. Instead of performing a true non-equilibrium transport simulation, this quantity can be extracted from the equilibrium fluctuations of the friction force in an equilibrium simulation by using a Green–Kubo relation[39,40]

$$\lambda = \lim_{t \to \infty} \lambda_{GK}(t) = \lim_{t \to \infty} \frac{1}{2Ak_B T} \int_0^t \langle \mathbf{F}_p(t') \cdot \mathbf{F}_p(0) \rangle dt' \quad (1)$$

where $k_B$ is the Boltzmann constant and $T$ the simulation temperature.

The integrated autocorrelation function of the carbon atom force components acting parallel to the graphite sheets $\lambda_{GK}(t)$ for our three configurations is shown in Fig. 6e. The friction coefficient $\lambda$ can be extracted from the plateau region, which is reached after about 0.4 ps. The comparison between the three simulations reveals that oxidation (red and blue lines in Fig. 6e) and the change of the stacking from AA to AB (blue and green lines in Fig. 6e) significantly reduce the friction coefficient.

This result is corroborated by an analysis of the *xy*-profiles of the oxygen atom distribution above the carbon atoms. The probability distribution maps $p_o(x,y)$ shown in Fig. 6f were gathered by mapping the lateral position of all O atoms along the trajectory on the underlying graphene lattice. The plots for GIC-AA and ox-GIC($C_{30}^+$)-AA indicate the preference of the O atoms of the sulfuric acid species to reside above the center of the carbon hexagons. However, the corrugation in the probability map is significantly lower for the oxidized compared to the neutral system. The origin of this difference is a change in the preferred orientation of the sulfuric acid molecules upon oxidation (see Supplementary Notes 4 and 5). AB stacking even further reduces the corrugation in the probability distribution. Here, the reason is that in AB stacking there is always a carbon atom below and above a carbon hexagon. The probability map therefore resembles closely the superposition of two shifted maps from AA stacking (see Supplementary Note 6).

The probability distribution $p_o(x,y)$ was converted into a free energy surface via $\Delta A(x,y) = -k_B T \ln p_o(x,y)$ from which a line scan (see Fig. 6g) along the direction depicted by the arrows in Fig. 6f was generated. From these line scans it becomes obvious that oxidation and changing the stacking sequence from AA to AB significantly reduce the free energy barrier for the sulfuric acid molecules to move parallel to the graphite sheet. This result confirms our findings for the friction coefficient, as the leading term in the friction coefficient is proportional to the square of the corrugation of the potential energy surface felt by the sulfuric acid molecules in contact with the graphite sheets[41].

**Intercalation and GIC oxidation mechanism**. The results of our MD simulations show that the slippage of the sulfuric acid molecules between the graphite layers, which is the essential part of the intercalation process for large grains, is drastically eased if oxidation has taken place. This might be the decisive factor to explain the experimental observation that the presence of an oxidizing agent is necessary for successful intercalation to occur. Furthermore, the stacking order is identified to strongly influence friction. The observation that the friction coefficient is lowest for the most favorable AB stacking but increases for AA-stacked graphite is in excellent agreement with the experimental results about the intercalation ability of various graphite types. A high crystallinity (i.e., well-ordered AB and ABC stacking) combined with large grain sizes as in NG1 efficiently reduces the corrugation of the free energy surface for the slip of the sulfuric acid molecules between the graphite sheets and supports successful stage-1 GIC formation. In contrast, any deviation (inherent stacking faults, small grains) as in NG2 and TG leads to an increase in the free energy barrier and thus hinders the intercalation process.

Finally, our observations on the stability of oxo-species in graphite sulfate suggest an alternative oxidation mechanism. Since oxo-species do not remain attached to the graphite sheets up to an oxidation state of $C_{30}^+$, it is not necessary for the rather bulky oxidizing agents (ammonium persulfate) to enter the confined space between the graphite layers. The electron transfer from the conducting carbon sheets to the oxidizing agent in the oxidation process can take place at the outside of the graphite stack and charge balance is then maintained by a Grotthuss diffusion of positively charged protons through the hydrogen bond network of the sulfuric acid molecules out of the interlayer region (Fig. 1b). This process may even continue beyond an oxidation state of $C_{30}^+$ and oxo-species only attach to the carbon scaffold upon hydrolysis.

## Discussion

Experimental and theoretical results demonstrate that crystalline graphite with large grain size can act as a precursor for wet-chemically prepared graphene of high quality, as directly observed by HRTEM and quantified by statistical Raman spectroscopy. Friction of sulfuric acid confined between layers of graphite is significantly lowered with electronic oxidation, which initiates the intercalation process. Intercalation and oxidation reinforce each other: if intercalation is hindered by high friction, oxidation remains insufficient and the partial intercalation is reversible upon addition of water. Epoxides and hydroxyl groups are not stable at low oxidation levels, but become protonated and cleaved from the carbon lattice. Only at higher oxidation levels oxo-species remain attached to the graphite layers.

Moreover, friction of confined sulfuric acid is lowest for AB-stacked graphene layers. Therefore, it is plausible that natural graphite, bearing grain boundaries and irregular stacking orders, is not an equally potent precursor for graphene production. Along the same line, turbostratic graphite is not forming a proper GIC (low oxidation degree) and thus reverts back to graphite upon water addition. Sulfuric acid, however, intercalates graphite of high crystallinity with large grains upon oxidation. The oxidation proceeds very likely outside the layer confinement by electron transfer, and charge compensation is accompanied by proton diffusion, leading to $HSO_4^-$ enrichment between the layers (Fig. 1b).

The here presented route towards oxo-functionalized graphene and graphene on substrates or stabilized by surfactants in water can be expected to also lead to novel graphene derivatives with tailored functionality suitable for electronic or bio-sensing applications.

## Methods

**General methods**. Graphite grades NG1 (grade 2012) and TG were obtained from Asbury Carbon. NG2 was purchased from Future Carbon. Large flake graphite was obtained from NGS Trading & Consulting GmbH. Sulfuric acid, hydroiodic acid 57% in water (HI), trifluoroacetic acid (TFA), and ammonium persulfate were obtained from Sigma-Aldrich® and used as received. Doubly distilled water was obtained from Carl Roth GmbH and used as received. Methanol was distilled in a solvent circulation apparatus before use. Centrifugation was accomplished by a VWR Micro 1814 table centrifuge. Sonication was performed using a Sonorex Digital P10 from Bandelin. LB films were prepared using a LB mini trough from KSV NIMA. AFM imaging was performed using a NT-MDT Solver Pro. Raman spectra were recorded with LabRAM HR Evolution from Horiba Scientific using an 532 nm laser line. Spectra were processed as reported elsewhere[42]. UV–Vis spectroscopy was conducted on a Perkin Elmer—Lambda 1050.

**Synthesis of graphite sulfate**. Graphite (0.50 g; 42 mmol) was suspended in sulfuric acid (20 ml; 96–98%). Over a period of three days ammonium persulfate (1.50 g; 7 mmol) was added in small portions under stirring.

**Synthesis of oxo-$G_1$**. Graphite sulfate (400 mg) was centrifuged (30 min; 14,000×g) and the supernatant was removed. The graphite sulfate was freezed by liquid nitrogen and doubly distilled water (1.5 ml) was added. The mixture was cooled for 16 h to 8 °C. The reaction mixture was purified by centrifugation and redispersion in water (four times 60 min; 14,000×g) until the supernatant was of neutral pH. Finally, the carbon material was taken up in 1/1 volume mixture of water/methanol. The dispersion was sonicated in a bath sonicator for 2 min. The oxo-$G_1$ solution was obtained by centrifugation (5 min; 700×g) and collecting the supernatant. Small particles were removed by centrifugation (60 min, 14,000×g). Finally, larger particles were removed by centrifugation (10 min; 700×g) leaving oxo-$G_1$ in dispersion.

**Preparation of LB films**. The as-prepared oxo-$G_1$ solution was slowly spread dropwise on the aqueous subphase using a syringe. The solution was spread until the surface pressure reached about 1 mN m$^{-1}$. The barriers were closed (20 mm min$^{-1}$) until the surface pressure reached 5 mN m$^{-1}$. The substrate was then pulled up (2 mm min$^{-1}$) to deposit flakes of oxo-$G_1$ on Si/SiO$_2$ (300 nm) substrates.

**Theory**. The Car-Parrinello molecular dynamics simulations were carried out using the CPMD code[43], the Perdew-Burke-Ernzerhof PBE exchange-correlation functional[44], Vanderbilt ultrasoft pseudopotentials[45] and a plane wave basis set with kinetic energy cutoff of 30 Ry. To account for dispersion effects, the semi-empirical Grimme D2 correction was applied[46]. The MD simulations were carried out in the NVT ensemble using a Nose-Hoover thermostat set to 400 K in order to compensate for the known overstructuring of hydrogen-bonded liquids at room temperature in PBE[47]. An additional Nose-Hoover thermostat, a fictitious mass of 400 a.u. and a time step of 4 a.u. were used for the propagation of the electronic degrees of freedom. For obtaining converged results for the friction coefficient in the MD simulations, large supercells with lateral size of $10 \times 6\sqrt{3}$ graphite lattice constants (i.e., 240 carbon atoms) and trajectories of more than 70 ps of simulation time were required. The distance between the graphite layers was set to 7.98 Å as determined experimentally for stage-1 graphite sulfate[22–24]. The interlayer region was filled with 32 sulfuric acid and four water molecules for the neutral GIC simulations and eight hydrogen atoms were removed in the oxidized GIC($C_{30}^+$) configurations. The sulfuric acid to water ratio mimics the experimental sulfuric acid concentration of about 98%. The total number of molecules included in the interlayer region was determined by initial test runs monitoring the stress tensor, but turned out to be very close to the experimentally observed graphite sulfate composition of $(C_{24}^+(HSO_4^-)(H_2SO_4)_2)_n$,[19–21].

**HRTEM**. The TEM investigations were performed with a FEI Titan 80–300 transmission electron microscope fitted with a CEOS CETCOR third order spherical aberration corrector and a Gatan UltraScan 1000XP CCD camera. The microscope was operated at an electron acceleration voltage of 80 kV and the electron source extraction voltage was set to 2 kV reducing the primary electron energy distribution to 0.5 eV FWHM[48]. The image acquisition time was 2 s and the dose rate for the nominally 300,000× magnified images (Fig. 4) less than $1.6 \times 10^6$ e$^-$ nm$^{-2}$ s$^{-1}$. The diameter of irradiated area was approximately two times the camera width and the sample area imaged was changed after recording few micrographs with varying defocus; therefore, the maximum total dose per imaged area was about $10^8$ e$^-$ nm$^{-2}$, since the post-sample beam shutter was used. Vignettes were balanced using Gwyddion software.

**Data availability**. The data that support the findings of this study are available from the corresponding authors upon request.

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

## Acknowledgements

The authors thank the Deutsche Forschungsgemeinschaft (DFG, Collaborative Research Center SFB 953 "Synthetic Carbon Allotropes", project C1, and grand no. EI938/3-1) and the Graduate School Molecular Science (GSMS) for financial support. S.S. thanks the Fonds der Chemischen Industrie (FCI) for a Chemiefonds Fellowship. Computational resources were provided by RRZE.

## Author contributions

S.E. and B.M. supervised the project as scientific group leaders and principal investigators and worked out the concept. F.G. prepared functionalized graphene films and C.E.H., F.G. and P.R. characterized functionalized graphene samples and related graphene. S.S. carried out and analyzed the AIMD calculations. U.K. and F.B. characterized films of graphene flakes by HRTEM. S.S., C.E.H., F.B., U.K., B.M., and S.E. wrote the manuscript.

## Additional information

**Competing interests:** The authors declare no competing financial interests.

