## [Peer Review File · Nature Communications]

Reviewers' Comments:

Reviewer #1:

Remarks to the Author:

In this work, authors investigate the process of intercalation and subsequent partial oxidation of graphite to form so-called "oxo-functionalized graphene", the term introduced earlier by Eigler for this type of graphene derivatives. The authors investigate the role of the graphite source to show that only high-crystallinity graphite undergo efficient intercalation and subsequent delamination. The quality of as-obtained graphene is confirmed by statistical Raman microscopy. The authors further enhance the study by theoretical calculations, showing conditions for sulfuric acid intercalation between the graphite layers. They also demonstrate that the graphene structures comprising just few alcohol or epoxy groups in the intercalated conditions are not energetically favorable and tend to defunctionalize with formation of the intact charged graphene. Calculations further suggest that no physical intercalation of an oxidizing agent might be needed for the oxidation of the graphene planes, which occurs via electron and proton transfer. These findings add significant insights to our understanding of both intercalation and oxidation of graphite to potentially develop controllable routes for mass production of graphene and graphene derivatives by subjecting them to oxidative acidic media. The findings are novel, and will be interesting for researchers working in the field, and for a broader community. The work is performed on the high professional level and deserves publication in Nature Communications after addressing two minor questions.

1. The ultimate goal of the studies in this field is obtaining graphene in a high yield, suitable for industrial applications. The method's effectiveness can be measured by the yield of graphene, i.e. by the degree of exfoliation. The authors did not specify this critical parameter in the manuscript neither in main text, nor in the experimental section. Authors start with 50 mg of graphite precursor. The question is what part of this amount is transferred into the liquid supernatant phase, that was used for fabricating thin films? How much graphite remained as the precipitate? How the graphite/graphene in precipitate is different/similar to that in the supernatant? These are the critical questions needed to understand the insights of the process. Specifying the yields on each step of the processing would be very informative.

2. Authors show an optical picture of the fabricated film (Fig. 4d), and only one HRTEM image of a single graphene flake. From these data it is difficult to understand the morphology of the prepared products. A SEM images of the film, showing size and other structural details of constituting flakes, acquired at different magnifications, would greatly help readers in understanding the nature of the obtained products.

3. Since authors capable of making graphene films (Fig. 4d), can they make one more step forward, and fabricate a macroscopic continuous film, and measure its transparency and sheet resistance? This final question is only a friendly suggestion; no insist on this part.

Sincerely,
Ayrat Dimiev

Reviewer #2:

Remarks to the Author:

The manuscript by Seiler et al. reports the mechanism of sulphuric acid intercalation of graphite and its oxidation. Their studies show that the stacking defects and density of defects in graphite significantly affect the intercalation and subsequent oxidation reaction due to the changes in the atomic scale friction of the molecules intercalated between graphene layers. This study provides a detailed understanding of the formation process of the graphite intercalation compounds and its subsequent oxidation. The reported results are interesting and novel. However, I have several concerns and that need to be addressed before the final acceptance.

1. I believe the proposed title is misleading. According to the presented results, the friction of sulphuric acid molecules on graphene affects the intercalation efficiency and associated oxidation

reaction. Hence it is more appropriate to use "graphene oxide formation" rather than "graphene formation" in the title.

2. Presentation of the manuscript could be further improved for clarity of the presented results. The introduction is not concise and needs more emphasis on the novelty of this study.

3. The thorough understanding of the intercalation and oxidation of graphite seems more interesting and novel than the preparation of graphene films with lower defect density. The latter has already been addressed by several groups using different techniques.

4. Why did the authors call graphene oxide as oxo-functionalized graphene (Oxo-G1)? This terminology is confusing; better to use the widely used graphene oxide rather than oxo-functionalised graphene.

5. Three different types of graphite (two natural graphite and one turbostratic graphite) has been used for this study. From the Raman spectra shown in Fig. 3 it is apparent that the defect density in turbostratic graphite is larger than natural graphite (larger D peak). How the authors differentiated the effect of defects and stacking disorder on the intercalation process? It is hard to imagine that the stacking disorder has a huge influence on the intercalation process. Several groups have already shown the possibility of alkali metal intercalation of misoriented graphene layers and reduced graphene oxide layers. I believe a larger defect density in turbostratic graphite is hindering the intercalation.

6. It is not very clear about the accuracy of the estimation of the distance between defects in different grades of graphite by using Raman spectroscopy. The details of this estimation are missing in the manuscript. It has been known that the defect separation estimation by Raman spectroscopy technique is only valid for point defects. How the authors ruled out the contribution of D peak intensity from the grain boundaries and its influence on the estimation.

7. It is not very clear how the authors calculated the area of single layer graphene by G-peak intensity (Figure S2). Again, defect density estimation in graphene oxide/reduced graphene oxide layers from Raman intensity ratio is known to be inaccurate. Need further justification for the use of this technique.

Reviewer #3:

Remarks to the Author:

The authors have studied the formation of oxo-G1 (a functionalized graphene derivative) from graphite using a combination of experimental and theoretical techniques. Their experimental results show that the crystal quality of the initial graphite sample plays a key role in the formation of oxo-G1. In particular, formation and oxidation of the stage-1 graphite sulfate intermediate is observed to be easier in the case of high quality graphite with perfect stacking than in the case of more defective graphite. Ab initio molecular dynamics simulations provide insight into these observations by showing that interaction between intercalated sulfuric acid molecules and oxo-species on the graphite layers result in electronic oxidation of these layers. In turn, this oxidation has a significant effect on the diffusion of the intercalated sulfuric acid molecules by reducing the friction between the molecules and the graphite layers. The stacking of these layers has also an influence on the friction, which is lower in the case of perfect AB stacking in comparison to AA stacking, as observed experimentally.

Altogether I find these results of very good quality and high interest. I have only few minor comments/questions concerning the theoretical part.

- In the ab initio simulations, the oxidation of the graphite layers is found to be limited to one positive charge per 30 carbon atoms. While this is in good agreement with the experimentally observed degree of oxidation of stage-1 GIC, it would be interesting to understand in more detail the reason of this specific value, e.g. through analysis of the electronic energy levels.

- The authors should comment on the accuracy of their calculations for the friction coefficient. In

particular, a point of concern is that there seems to be no real plateau value for the red line in Figure 6e.

- The explored configurations of isolated pairs of hydroxyl groups in Figure 5 are not very realistic. Analysis of more realistic configurations would be desirable, in particular configurations involving combined hydroxyls and epoxides, as discussed extensively in the literature.

Reviewers' comments:

Reviewer #1 (Remarks to the Author):

In this work, authors investigate the process of intercalation and subsequent partial oxidation of graphite to form so-called "oxo-functionalized graphene", the term introduced earlier by Eigler for this type of graphene derivatives. The authors investigate the role of the graphite source to show that only high-crystallinity graphite undergo efficient intercalation and subsequent delamination. The quality of as-obtained graphene is confirmed by statistical Raman microscopy. The authors further enhance the study by theoretical calculations, showing conditions for sulfuric acid intercalation between the graphite layers. They also demonstrate that the graphene structures comprising just few alcohol or epoxy groups in the intercalated conditions are not energetically favorable and tend to defunctionalize with formation of the intact charged graphene. Calculations further suggest that no physical intercalation of an oxidizing agent might be needed for the oxidation of the graphene planes, which occurs via electron and proton transfer.

These findings add significant insights to our understanding of both intercalation and oxidation of graphite to potentially develop controllable routes for mass production of graphene and graphene derivatives by subjecting them to oxidative acidic media. The findings are novel, and will be interesting for researchers working in the field, and for a broader community. The work is performed on the high professional level and deserves publication in Nature Communications after addressing two minor questions.

Answer: We very much appreciate the high rating of our manuscript and work by the reviewer.

1. The ultimate goal of the studies in this field is obtaining graphene in a high yield, suitable for industrial applications. The method's effectiveness can be measured by the yield of graphene, i.e. by the degree of exfoliation. The authors did not specify this critical parameter in the manuscript neither in main text, nor in the experimental section. Authors start with 50 mg of graphite precursor. The question is what part of this amount is transferred into the liquid supernatant phase, that was used for fabricating thin films? How much graphite remained as the precipitate? How the graphite\graphene in precipitate is different\similar to that in the supernatant? These are the critical questions needed to understand the insights of the process. Specifying the yields on each step of the processing would be very informative.

Answer: We agree with the reviewer that high yield and high quality are the key for industrial scale production. Here, we demonstrate a high quality by understanding the intercalation process. Up to now we must admit that a high yield process is not available. We did not focus on the yield of the process yet, but demonstrate for the first time that defects in the carbon lattice can be avoided and films of single layer flakes can be produced. Here, we usually end up with a 3 mL vial of single layer of functionalized graphene of the demonstrated quality. Such a vial is then used for making Langmuir-Blodgett films of the demonstrated quality.

Since we believe, up to now it is not possible to give a yield in terms of %, we made this point clear in the manuscript and clarified that the centrifugation process finally yields a

dispersion of functionalized graphene with an optical density of about 0.5, suitable to prepare Langmuir-Blodgett films even after dilution by 1:10 (however, not shown in the manuscript). In the manuscript we added:

“Although a larger fraction of the functionalized material remains few- and multi-layered, dispersions of oxo-G₁ with an optical density of 0.5 could be obtained by iterative centrifugation and redispersion in a 1/1 mixture of H₂O/MeOH assisted by ultrasonication (Supplementary Fig. 1).”

The experience gained by the experiments with that material tells us that the precipitate can still be used to isolate more single layers of the same quality and the analysis of few-layers, which precipitate show a highly intact carbon lattice. For statistical Raman analysis based on single layers of graphene, we filtered out the minor fraction of few-layer graphene and multi-layer graphene. The I_D/I_G ratio of few-layer graphene was determined to about 0.75 ± 0.2 and the FWHM of the 2D band is 44 cm⁻¹ +/- 7 cm⁻¹. For multi-layer graphene the I_D/I_G ratio is only 0.42 ± 0.1 and FWHM 2D is 47 +/- 7 cm⁻¹. We included the new **Supplementary Fig. 3** (together with a reference in the manuscript to the new supplementary material) showing the detailed statistical Raman analysis of few-layers (filtered by detector counts of the I_G band; between 540 and 1200 detector counts) and multi-layers (> 1200 detector counts). We included also spectra related to few-layer graphene and multi-layer graphene. Therefore, we conclude that functionalization takes also place in few-layers, but the yield of delamination, formation of single layers, remains a future challenge, in particular using volatile solvents, such as water and methanol, avoiding high boiling point solvents, such as N-methylpyrrolidone, which is difficult to remove from the surface and impossible at room temperature.

2. Authors show an optical picture of the fabricated film (Fig. 4d), and only one HRTEM image of a single graphene flake. From these data it is difficult to understand the morphology of the prepared products. A SEM images of the film, showing size and other structural details of constituting flakes, acquired at different magnifications, would greatly help readers in understanding the nature of the obtained products.

Answer: We are aware that the morphology of the Langmuir-Blodgett film is of particular interest. Thus, the manuscript includes an optical overview micrograph in **Fig. 4d** showing the single layer coverage of the SiO₂/Si wafer by the intermediate contrast. We improved the figure that single layers are identified easier. Moreover in **Fig. 4b** we included an AFM image that shows the typical size of single layer flakes. The diameter of many flakes is around 2 μm. In addition, we added overview HRTEM micrographs in the new **Supplementary Fig. 5** of randomly selected places, with the diffraction patterns included, and a reference to this new supplementary material was included in the manuscript.

3. Since authors capable of making graphene films (Fig. 4d), can they make one more step forward, and fabricate a macroscopic continuous film, and measure its transparency and sheet resistance? This final question is only a friendly suggestion; no insist on this part.

Answer: We agree that this information would be of high value. Up to now we can only give an estimation. Since 1-2 layers are mainly present, the transparency may be around 95%.

Due to the high quality of graphene, the sheet resistance will depend on the doping of graphene. In Adv. Mater. 25 (2013) 3583 we demonstrated the quality of single flakes. The focus of our research is currently on the flake to flake contact and the measurement of physical properties of overlapping flakes. Thus, we agree with the reviewer that transparency and sheet resistance are highly important and we will address these issues in our current and future work.

Reviewer #2 (Remarks to the Author):

The manuscript by Seiler et al. reports the mechanism of sulphuric acid intercalation of graphite and its oxidation. Their studies show that the stacking defects and density of defects in graphite significantly affect the intercalation and subsequent oxidation reaction due to the changes in the atomic scale friction of the molecules intercalated between graphene layers. This study provides a detailed understanding of the formation process of the graphite intercalation compounds and its subsequent oxidation. The reported results are interesting and novel. However, I have several concerns and that need to be addressed before the final acceptance.

Answer: We thank the reviewer for pointing out the novelty of our manuscript.

1. I believe the proposed title is misleading. According to the presented results, the friction of sulphuric acid molecules on graphene affects the intercalation efficiency and associated oxidation reaction. Hence it is more appropriate to use “graphene oxide formation”; rather than “graphene formation” in the title.

Answer: Using graphene oxide formation could be used in the title, since we describe formation of oxo-groups on the surface of graphene. However, using the term graphene oxide would also be misleading, since graphene oxide is a class of compounds that is associated by a ruptured hexagonal lattice of carbon atoms with various functionality at the edges of defects, a rearranged carbon lattice and an overall degree of functionalization of around 50%. Here, we describe that the hexagonal lattice remains by far stable, while oxo-functional groups are formed on the hexagonal basal plane (see also our answer to Remark 4). However, the reviewer is right that addition of oxo-addends to the lattice is the process, which we mainly describe. In a second step the oxo-groups are cleaved by reduction to form graphene. Thus, we changed the title to: “Friction on the nanoscale as key to oxidative graphite intercalation and high-quality graphene formation”.

2. Presentation of the manuscript could be further improved for clarity of the presented results. The introduction is not concise and needs more emphasis on the novelty of this study.

Answer: The reviewer is right, the introduction covers the context of this research, but a description pointing on the major novelty comes quite late. Thus, we re-organized our introduction and re-wrote a large part of it – see yellow marked region in the manuscript file.

3. The thorough understanding of the intercalation and oxidation of graphite seems more interesting and novel than the preparation of graphene films with lower defect density. The latter has already been addressed by several groups using different techniques.

Answer: We agree with reviewer that understanding the intercalation process and oxidation is of utmost importance. The formation of graphene is a supplement, however, graphene films of the here presented quality have not yet demonstrated. E.g. Raman spectra with FWHM of the 2D peak as narrow as around 30 cm^{-1} is rare, in particular for a whole film. Many examples in the literature are available that display films of flakes and I_D/I_G ratios of Raman spectra that are even lower than around 1. However, 2D peaks are often much broader than around 40 cm^{-1} and such presented spectra stem from few-layer graphene, rather than single layer graphene. In addition HRTEM micrographs with atomic resolution and without pretreatment of the materials are also rare. However, although we feel that we achieved significant improvement, we agree that the focus of the work is on the friction of sulfuric acid, which enables functionalization and in last consequence the formation of films of flakes of graphene.

4. Why did the authors call graphene oxide as oxo-functionalized graphene (Oxo- G_1)? This terminology is confusing; better to use the widely used graphene oxide rather than oxo-functionalised graphene.

Answer: The reason, why we discriminate from the term graphene oxide is very simple. Graphene oxide is a collective term that includes any type of single layers of carbon (not necessarily hexagonal) with various types of oxo-functionality on top of the basal plane and at the rims of defects. The latter defects contribution is that high, that even reduction leads not to graphene, but a ruptured and rearranged carbon lattice that may even not possess a hexagonal long-range order, as demonstrated recently [ACS Nano 10 (2016) 7515–7522; DOI: 10.1021/acsnano.6b02391].

In contrast, we emphasize on the single layer nature of oxo-functionalized graphene and consider the oxo-functionality as an addend on both sides of the basal plane (similar to addends on fullerenes). Chemical reduction can subsequently quantitatively remove the oxo-functionality from the carbon basal plane, forming graphene (see Raman investigation).

We now emphasized more on the difference between oxo- G_1 and graphene oxide in our newly written introducing section: “This oxo-functionalized graphene was found to be suitable to establish a controlled chemistry of graphene where lattice defects play a minor role.^{12, 15-18} We term this product oxo- G_1 (index = number of layers) in order to distinguish it from the known graphene oxide, which possesses an undefined amount of defects and consequently a disturbed order of the lattice.”

5. Three different types of graphite (two natural graphite and one turbostratic graphite) have been used for this study. From the Raman spectra shown in Fig. 3 it is apparent that the defect density in turbostratic graphite is larger than natural graphite (larger D peak). How the authors differentiated the effect of defects and stacking disorder on the intercalation

process? It is hard to imagine that the stacking disorder has a huge influence on the intercalation process. Several groups have already shown the possibility of alkali metal intercalation of misoriented graphene layers and reduced graphene oxide layers. I believe a larger defect density in turbostratic graphite is hindering the intercalation.

Answer: Thank you for this comment. The question aims on the effect of stacking order vs. in-plane lattice defects on the intercalation process. Indeed we do not rule out that lattice defects play a role. However, our results suggest that in-plane lattice defects play a minor role, although one can speculate about kinetic effects. Regarding the D-peak intensity in natural graphite vs. turbostratic graphite: Generally speaking the I_D/I_G ratio is 0.1 for natural graphite and 0.4 for turbostratic graphite. Assuming point defects, those spectroscopic data relate to distances of defects of 35 nm for natural graphite and 19 nm for turbostratic graphite (also described and referenced in the manuscript). A general difference in defects of 35 nm and 20 nm respectively would not be a significant reason to describe the differences in intercalation behavior. In addition grain boundaries are also present in any graphite and those may be barriers for intercalations. However, since the laser spot of the Raman laser is much larger than the grain boundary width and since e.g. in natural graphite the 3R-phase has crystallite sizes larger than 10 μm , the contribution to the D peak is assumed to be small. For graphites with smaller crystallite phase, such as 100 nm, the grains are still larger than the extracted distance of point defects, according to the used model. An interesting Raman study on point defects and line defects can be found here: 2D Mater. 4 (2017) 025039; DOI: 10.1088/2053-1583/aa5e77. However, HRTEM micrographs give no evidence for line defects.

It is important to recall, what we discussed in the manuscript: Modeling of large systems, which include defects, on the level of theory presented here, is not possible (the exact shape and dimension is not well known). However, here we show that the stacking order has already a strong influence on the friction of sulfuric acid confined in the interlayer galleries and thus, grain boundaries and certainly also lattice defects, all together, can be assumed to hinder the ease of intercalation. We propose, that a graphite with a large crystal size of 3R-phase would be an ideal starting material for preparing a high quality of graphene with increased yield.

6. It is not very clear about the accuracy of the estimation of the distance between defects in different grades of graphite by using Raman spectroscopy. The details of this estimation are missing in the manuscript. It has been known that the defect separation estimation by Raman spectroscopy technique is only valid for point defects. How the authors ruled out the contribution of D peak intensity from the grain boundaries and its influence on the estimation.

Answer: As we already discussed in the answer of point 5 we estimate that the influence of grain boundaries on the intensity ratio of the D and G peak is minor. In contrast, point defects in graphite are well documented and HRTEM micrographs visualized such point defects in our samples. In contrast, we do not find line defects. They would be visible in overview micrographs (HRTEM, **Supplementary Fig. 5**) The influence of line defects on the I_D/I_G ratio was recently demonstrated [2D Mater. 4 (2017) 025039; DOI: 10.1088/2053-1583/aa5e77]. Our own investigations, also conducted on graphene prepared from different sources, reveal that the model introduced by Cancado and Lucchese (see manuscript) fits

quite well to the observations made by HRTEM (see e.g. Angew. Chem. Int. Ed. 56 (2017) 9222; DOI: 10.1002/anie.201704419).

7. It is not very clear how the authors calculated the area of single layer graphene by G-peak intensity (Figure S2). Again, defect density estimation in graphene oxide/reduced graphene oxide layers from Raman intensity ratio is known to be inaccurate. Need further justification for the use of this technique.

Answer: In first approximation, the G-peak intensity depends on the number of probed carbon atoms. Thus, within experimental errors the intensity of the G peak relates to single layers, bilayers or few-layers. We elaborated this technique in detail and used the spectroscopic information to image the quality and shape of flakes (Raman microscopy). Moreover, the Raman results were correlated to AFM results. By this approach, we proved that the Raman intensity and determination of the single layer content can indeed be related (see: J. Phys. Chem. C 118 (2014) 7698-7704; DOI: 10.1021/jp500580g).

Reviewer #3 (Remarks to the Author):

The authors have studied the formation of oxo-G₁ (a functionalized graphene derivative) from graphite using a combination of experimental and theoretical techniques. Their experimental results show that the crystal quality of the initial graphite sample plays a key role in the formation of oxo-G₁. In particular, formation and oxidation of the stage-1 graphite sulfate intermediate is observed to be easier in the case of high quality graphite with perfect stacking than in the case of more defective graphite. *Ab initio* molecular dynamics simulations provide insight into these observations by showing that interaction between intercalated sulfuric acid molecules and oxo-species on the graphite layers result in electronic oxidation of these layers. In turn, this oxidation has a significant effect on the diffusion of the intercalated sulfuric acid molecules by reducing the friction between the molecules and the graphite layers. The stacking of these layers has also an influence on the friction, which is lower in the case of perfect AB stacking in comparison to AA stacking, as observed experimentally.

Altogether I find these results of very good quality and high interest. I have only few minor comments/questions concerning the theoretical part.

1. In the *ab initio* simulations, the oxidation of the graphite layers is found to be limited to one positive charge per 30 carbon atoms. While this is in good agreement with the experimentally observed degree of oxidation of stage-1 GIC, it would be interesting to understand in more detail the reason of this specific value, e.g. through analysis of the electronic energy levels.

Answer: We would like to emphasize first that the oxidation state of about one positive charge per 30 carbon atoms in stage-1 GIC is only the limit up to which attached oxo-species like OH groups and epoxides are unstable and are immediately protonated by the sulfuric acid and detach as water molecules/H₃O⁺ from the graphite sheets. Of course, with stronger

oxidizing agents, charge states exceeding C_{30}^+ can be obtained, but then oxo-species will remain attached on the graphite layers.

We agree with the reviewer that a more detailed understanding of why the stability of the oxo-species changes at an oxidation state of about C_{30}^+ would be very interesting and we thank her/him for this suggestion. To this aim we have calculated projected density of states (DOS) from representative snapshots of our simulations for non-oxidized and oxidized GIC. This data is shown in the new **Supplementary Note 2** and a reference to this new supplementary material is given in the manuscript. In addition, the iso-electron density contour surfaces of states integrated up to approximately 2 and 4 electrons below the respective Fermi levels are included.

The DOS plots show that oxidizing the stage-1 GIC to C_{30}^+ leads to a downward shift of the Fermi level of about 1 eV. The states, which are depopulated by removing 2 electrons from a unit cell with 60 carbon atoms, are the π -states of the carbon scaffold (see contour plots on the left side in **Supplementary Fig. 7**). For the oxidized GIC- C_{30}^+ , however, O states of the sulfuric acid molecules are already close below the Fermi level. Thus, going to higher oxidation states than C_{30}^+ by removing additional electrons would not only depopulate carbon states but also O states of the sulfuric acid molecules, corresponding to a partial oxidation of the sulfuric acid (see right side in **Supplementary Fig. 7**). Therefore, at these higher oxidation states it becomes more favorable that oxo-species remain attached on the graphite scaffold.

2. The authors should comment on the accuracy of their calculations for the friction coefficient. In particular, a point of concern is that there seems to be no real plateau value for the red line in Figure 6e.

Answer: The evaluation of the Green-Kubo formula only leads to a true constant plateau if the thermodynamic limit is reached, *i.e.*, for infinite system size and infinite sampling time [Bocquet and Barrat, Phys. Rev. E 49 (1994) 3079]. Consequently, any finite system size simulation suffers from this issue and in practice yields an autocorrelation function with a maximum and a small plateau at short simulation times (typically a time on the order of the initial, fast relaxation of the system) which then decays to zero. This onset of the decline to zero can already be seen for the red line in **Fig. 6e**. This problem is particularly challenging for *ab initio* MD simulations with their limitations in system size and simulation time due to the high computational cost. However, if system size and required simulation time are carefully checked, converged friction coefficients can be extracted already from the first initial maximum with its small plateau [Bocquet *et al.*, J. Stat. Phys. 89 (1997) 321-346]. Tests for the convergence of the GK friction coefficient with system size and simulation time are usually done by comparing to classical MD simulations where longer simulations for larger unit cells can easily be done (see, for example, Falk *et al.*, Nano Letters 10 (2010) 4067-4073; Tocci, Joly and Michaelides, Nano Letters 14 (2014) 6872-6877). We also performed such tests with a classical force field, which showed that we had to use a rather large (10×6×3) graphene unit cell (*i.e.*, 240 atoms) to obtain converged results. Fortunately for the simulation time it turned out that about 40 ps are sufficient. A similar value was also reported in the *ab initio* study of Tocci *et al.*

A second important test for checking the reliability of the calculated friction coefficients is to assess the statistical error in the auto-correlation function. Here we followed the approach of Friedberg and Cameron [J. Chem. Phys 52 (1970) 6049], which is summarized in the book

“Computer Simulations of Liquids” from Allen and Tildesley. Briefly, from a block analysis the statistical inefficiency parameter is extracted which describes after how many time steps the MD trajectory provides uncorrelated, new information for the quantity of interest. For the force auto-correlation function from our MD simulations we found a value for the statistical inefficiency of about 1000 MD time steps. Thus, new starting points for calculating the force autocorrelation function were set at every 1000th MD step. The variance in the mean within this block averaging is taken as statistical error and is shown by the shaded lines in **Fig. 6e** of the manuscript. Thus, the onset of the decline of the red curve from its plateau value is still well within the margins of the statistical uncertainty.

To confirm the reliability of our friction coefficient calculations we have added our convergence test with simulation time and our assessment of the statistical error as new **Supplementary Note 3**.

3. The explored configurations of isolated pairs of hydroxyl groups in Figure 5 are not very realistic. Analysis of more realistic configurations would be desirable, in particular configurations involving combined hydroxyls and epoxides, as discussed extensively in the literature.

Answer: Our DFT calculations as well as previous studies (*e.g.*, Phys. Rev. Lett. 103 (2009) 086802) show that pairs of OH groups have a very strong preference to sit on opposite sides of a graphene sheet on a neighboring pair of carbon atoms (1,2 trans motif). Therefore, we started to build our model structures by combining such OH pairs. But of course, the reviewer is right that such OH pairs tend to cluster and mix with epoxides. Although we have focused on OH groups in our simulations, we do not want to exclude the existence of epoxy groups in the prepared oxo-functionalized graphene derivatives. However, we have to emphasize the fact that the degree of functionalization in the experiments is only around 4%, as we found and cited in our earlier work [Chem. Commun. 51 (2015) 3162]. For comparison, graphene oxide, prepared by standard methods, possesses a degree of functionalization of around 50%. Therefore, to mimic this low and dilute degree of oxo-functionalization in experiment, we decided to distribute our pairs of OH groups and not cluster them together, since the energy difference between clustering and diluting the OH pairs is much smaller than the energy cost for separating the OH pairs. These structures only served as simple model systems to study the impact of progressing oxidation. We also performed simulations with a quarter and with a half monolayer of OH and epoxy groups. In all cases we observed the spontaneous transfer of only two to three protons from the sulfuric acid molecules to OH or epoxy species (for a graphene unit cell with 60 atoms) during the simulation time, which always exceeded 15 ps. The protonated OH groups finally desorbed from the graphene sheet and moved into the liquid.

Reviewers' Comments:

Reviewer #1:

Remarks to the Author:

I support the publication. No further revision is needed.

Reviewer #2:

Remarks to the Author:

I believe the author's comments about the conventional graphene oxide (GO) is controversial. For example "Conventional synthesis of graphene via graphite intercalation, oxidation, delamination into graphene oxide and subsequent reduction (see Figure 1a for a schematic illustration of the process), however, leads to a ruptured and amorphous carbon lattice." The quality of GO is entirely depended on the detailed method of preparation, and it is tough to generalise the statement about its structure. One can easily find other electron microscopy references (diffraction and high-resolution electron microscopy) where long-range order and high crystallinity of pristine GO is demonstrated. However, I believe this is not a critical point of this paper, and the authors addressed my all other comments sufficiently. I do not have any further comments and support publication.

Reviewer #3:

Remarks to the Author:

The authors have satisfied my concerns. I am pleased with the additional theoretical analysis that has been provided both in the response letter and in the revised supporting material.

Reviewers' comments:

Reviewer #1 (Remarks to the Author):

I support the publication. No further revision is needed.

Reviewer #2 (Remarks to the Author):

I believe the author's comments about the conventional graphene oxide (GO) is controversial. For example "Conventional synthesis of graphene via graphite intercalation, oxidation, delamination into graphene oxide and subsequent reduction (see Figure 1a for a schematic illustration of the process), however, leads to a ruptured and amorphous carbon lattice". The quality of GO is entirely depended on the detailed method of preparation, and it is tough to generalise the statement about its structure. One can easily find other electron microscopy references (diffraction and high-resolution electron microscopy) where long-range order and high crystallinity of pristine GO is demonstrated. However, I believe this is not a critical point of this paper, and the authors addressed my all other comments sufficiently. I do not have any further comments and support publication.

Answer: We thank the reviewer for his comments about the too general statement on the structure of graphene oxide (GO). We agree that GO can possess a more or less damaged carbon framework. The degradation of the carbon framework during GO formation is a continual process and depends on reaction conditions. At the lower end of this range, the carbon lattice is amorphous, although sheets are still present. This lower end is indicated in the cited literature. Certainly, there are also other types of GO of higher quality. However, GO preparation by *standard* protocols leads to a ruptured carbon lattice, as indicated by Raman spectroscopy. Therefore, GO after reduction does not show the properties of pristine graphene. Accounting for the complexity and wide range of quality of graphene oxide materials, we changed the sentence in the manuscript, which now ends with "leads in most cases to a ruptured carbon lattice."

Reviewer #3 (Remarks to the Author):

The authors have satisfied my concerns. I am pleased with the additional theoretical analysis that has been provided both in the response letter and in the revised supporting material.